

# An Atlantic streamer in stratospheric ozone observations and SD-WACCM simulation data

Klemens Hocke[1,2], Franziska Schranz[1], Eliane Maillard Barras[3], Lorena Moreira[1,2], and Niklaus Kämpfer[1,2]

[1]Institute of Applied Physics, University of Bern, Bern, Switzerland
[2]Oeschger Centre for Climate Change Research, University of Bern, Bern, Switzerland
[3]Federal Office of Meteorology and Climatology, MeteoSwiss, Payerne, Switzerland

*Correspondence to:* K. Hocke
(klemens.hocke@iap.unibe.ch)

**Abstract.** Observation and simulation of individual ozone streamers are important for the description and understanding of nonlinear transport processes in the middle atmosphere. A sudden increase in mid-stratospheric ozone occurred above Central Europe on December 4, 2015. The GROunbased Millimeter-wave Ozone Spectrometer (GROMOS) and the Stratospheric Ozone MOnitoring RAdiometer (SOMORA) in Switzerland measured an ozone enhancement of about 30% at 34 km altitude
from December 1 to December 4. A similar ozone increase is simulated by the Specified Dynamics-Whole Atmosphere Community Climate (SD-WACCM) model. Further, the global ozone fields at 34 km altitude from SD-WACCM and the satellite experiment Aura/MLS show a remarkable agreement for the location and the timing of an ozone streamer (large-scale tongue like structure) extending from the subtropics in Northern America over the Atlantic to Central Europe. This agreement indicates that SD-WACCM can inform us about the wind inside the Atlantic ozone streamer. SD-WACCM shows an eastward wind of
about 100 m/s inside the Atlantic streamer in the mid-stratosphere. SD-WACCM shows that the Atlantic streamer flows along the edge region of the polar vortex. The Atlantic streamer turns southward at an erosion region of the polar vortex located above the Caspian Sea. The spatial distribution of stratospheric water vapour indicates a filament outgoing from this erosion region. The Atlantic streamer, the polar vortex erosion region and the water vapour filament belong to the process of planetary wave breaking in the so-called surf zone of the Northern mid-latitude winter stratosphere.

## 1 Introduction


Rossby wave breaking contributes to the mean meridional circulation and to the horizontal mixing of tropical, subtropical and extratropical air masses in the middle atmosphere (Waugh, 1996; Randel et al., 1993; Leovy et al., 1985). Rossby wave breaking occurs in the mid- and upper stratosphere during the winter season. Particularly the mid-latitudes are regarded as the surf zone of the stratosphere where the material erosion of the polar vortex takes place (McIntyre and Palmer, 1984). The
material erosion of the vortex leads to water vapour filaments at mid-latitudes since the vortex air is rich in water vapour which has a long life-time in the stratosphere. Müller et al. (2003) utilized simulation and observation data of stratospheric water vapour as a tracer for vortex filamentation in the Arctic winter.



Strong planetary waves shift the stratospheric polar vortex equatorwards, and subtropical air is drawn in tongue-like structures (streamers) from the subtropics to the extratropics. Offermann et al. (1999) observed the formation of the so-called Atlantic streamer in the trace gases $N_2O$ and $HNO_3$. Krüger et al. (2005) derived climatological features of stratospheric streamers by means of the FUB-CMAM model with increased horizontal resolution ($2.8° \times 2.8°$). They found that tropical-subtropical streamers mainly occur over the Atlantic and the East Asia/West Pacific region during Arctic winter. They emphasized that stratospheric streamers have nothing to do with ozone laminae (small-scale structures in vertical space) in the lower stratosphere. Khosrawi et al. (2005) reproduced the streamer distribution observed by the CRISTA experiment onboard Space Shuttle (Offermann et al., 1999) with the Chemical Lagrangian Model of the Stratosphere (CLaMS) and the Karlsruhe Simulation Model of the Middle Atmosphere (KASIMA). These model-observation intercomparisons indicate that planetary wave breaking and its induced stratospheric streamers are an excellent test for nonlinear wave-mean flow interactions in middle atmospheric chemistry-climate models.

In the following, we investigate whether the Specified Dynamics Whole Atmosphere Community Climate Model (SD-WACCM) can simulate an individual Atlantic streamer event which was observed by the GROunbased Millimeter-wave Ozone Spectrometer (GROMOS) at Bern and the satellite experiment Aura Microwave Limb Sounder (Aura/MLS). Further, we look in detail on the role of the Atlantic streamer in the process of planetary wave breaking and polar vortex erosion.

## 2 Data sets

### 2.1 The microwave radiometers GROMOS and SOMORA

The study is partly based on stratospheric ozone profiles observed by the GROund-based Millimeter-wave Ozone Spectrometer GROMOS and the Stratospheric Ozone MOnitoring RAdiometer SOMORA. The instruments are ground-based ozone microwave radiometers which are part of the Network for the Detection of Atmospheric Composition Change (NDACC). They continuously observe the middle atmosphere above Bern, Switzerland (46.95°N, 7.44°E, 577 m above sea level) and above Payerne, Switzerland (46.82°N, 6.95°E, 471 m above sea level). While the routine observations of GROMOS started in 1994, SOMORA measures since the year 2000. Both radiometers measure the thermal microwave emission of a rotational transition of ozone at 142.175 GHz. In our study, we use ozone profiles with an integration time of 2 hours for GROMOS and 1 hour for SOMORA. The hourly ozone profiles of SOMORA are averaged with a 3 hour-running mean in order to get close to the 2-hourly data of GROMOS and SD-WACCM. The altitude range of the ozone profiles is from 25 to 70 km with a vertical resolution of about 12 km in the stratosphere. The measurement response between 50 and 0.5 hPa (20 to 52 km) is higher than 0.8 (corresponding with an a priori contribution less than 20%). Therefore, the retrieved ozone values at these altitudes are primarily based on the measured line spectrum. For technical details, measurement principle and retrieval procedure of the instruments, see for example Moreira et al. (2015), Peter (1997), Hocke et al. (2007), Maillard Barras et al. (2009, 2015) and references included therein. An intercomparison study of Hocke et al. (2007) indicated that the relative differences between SOMORA and Aura/MLS are less than 10%. Similar values of uncertainty are obtained for GROMOS. The SOMORA instrument is quite similar to GROMOS and was also upgraded with a FFT spectrometer in 2009. The vertical ozone profiles



from GROMOS and SOMORA have been validated by means of nearby ozone sondes, ground stations and collocated satellite measurements, the data sets have been used for studies of ozone-climate interaction, middle atmospheric dynamics as well as for long-term monitoring of the stratospheric ozone layer and for detection of trends (Moreira et al., 2015; Studer et al., 2014, 2012; Keckhut et al., 2010; Flury et al., 2009; Steinbrecht et al., 2009; Hocke et al., 2007; Dumitru et al., 2006).

## 2.2 The Aura Microwave Limb Sounder

The Microwave Limb Sounder is an instrument onboard the NASA Aura satellite which was launched in July 2004. The level2 data of Aura/MLS consist of atmospheric vertical profiles with a spacing of 165 km (1.5° along the satellite orbit which is sun-synchronous with an inclination of 98° and a period of 98.8 min (Waters et al., 2006; Schwartz et al., 2008). This relatively dense, horizontal sampling should be sufficient for observation of ozone streamers. The vertical resolution of the ozone profiles of Aura/MLS ranges from 3 km in the stratosphere to 6 km in the mesosphere (Schwartz et al., 2008). The present study utilizes Aura/MLS data of the version 4.2. The global ozone maps of Aura/MLS were computed by interpolating the valid ozone profiles of one day to a horizontal grid ($2° \times 2°$) using a Delaunay triangulation (Matlab function TriScatteredInterp.m).

## 2.3 The SD-WACCM model

The Specified Dynamics-Whole Atmosphere Community Climate Model (SD-WACCM) was described and evaluated in detail by Brakebusch et al. (2013). Here, we use the Community Earth Systen Model (CESM) version 1.2.2 WACCM component set which is a coupled chemistry climate model of the National Center for Atmospheric Research (NCAR). The WACCM chemistry module is taken from the Model for OZone And Related chemical Tracers (MOZART) (Brasseur et al., 1998) but is extended to include 122 species (Lamarque et al., 2012). SD-WACCM is a modified version of WACCM in which the meteorology is constrained to match observations to within a user-defined tolerance (Lamarque et al., 2012; Kunz et al., 2011; Brakebusch et al., 2013). SD-WACCM is nudged with winds, temperature, surface pressure, surface wind stress and heat fluxes from the Goddard Earth Observing System 5 (GEOS5) analysis (Rienecker et al., 2008). The nudging coefficient is in our study 0.1, i.e., the winds, temperature and surface pressure are defined by a linear combination of 10% from GEOS5 and 90% from the model. Nudging is applied every 30 min. The model run was initialised on July 1, 2015 by means of a former WACCM run and GEOS5 data. The model output files are written every two hours, the horizontal resolution is $1.9° \times 2.5°$ (latitude $\times$ longitude), and the vertical resolution is about 1km in the stratosphere. The altitude range of SD-WACCM is from the surface to 140 km whereby nudging is only applied below 50 km. SD-WACCM can resolve planetary waves while short-term gravity waves are parameterized (Brasseur et al., 1998).

## 3 Results

The initial point of the present study was the occurrence of an ozone peak in mid-stratospheric ozone at Bern on December 4, 2015. Figure 1a) shows the time series of ozone at 34 km altitude above Bern as observed by the GROMOS microwave radiometer at Bern and the SOMORA microwave radiometer at Payerne. Ozone suddenly increases by about 30% from De-





cember 1 to December 4. The time series of ozone from SD-WACCM is shown in Figure 1b). Generally the ozone time series in Figure 1b) is smoother than in Figure 1a). SD-WACCM also sees the increase in ozone around December 4, 2015. The maximum in zonal wind of SD-WACCM (Figure 1c) is also at the time of the ozone peak observed by GROMOS (Figure 1a). In summary, ozone-rich air passed with a velocity of about 90 m/s at 34 km altitude above Bern.

The next step is to derive the vertical ozone profiles of GROMOS, SD-WACCM, and Aura/MLS above or close to Bern which are shown on the left-hand-side of Figure 2 for December 1, 2015 (dashed lines) and December 4, 2015 (solid lines). Ozone reaches a maximum of 9 ppm at 37 km altitude in case of GROMOS and a maximum of about 8 ppm at the same altitude in case of SD-WACCM and Aura/MLS. The ozone increase takes place in the mid-stratosphere between 30 and 45 km. This layer thickness of 15 km is given by the full width at half maximum of the ozone peaks of the difference profiles of
SD-WACCM, Aura/MLS and GROMOS at the right-hand-side of Figure 2. The double peak-structure in the difference profile of SD-WACCM is confirmed by the Aura/MLS observation. The vertical resolution of GROMOS (12 km) is not sufficient for resolving such a double peak. The profiles of Aura/MLS and SD-WACCM are not folded with the averaging kernels of GROMOS since we do not like to degrade the vertical resolution of the ozone profiles of Aura/MLS and SD-WACCM.

    The formation and the decay of the Atlantic streamer is shown in Figure 3a), b) and c) which show the global ozone field at
34 km altitude as simulated by SD-WACCM for December 1, 2015, December 4, 2015 and December 8, 2015, respectively. We selected the polar stereographic projection in order to be in the same situation as an observer in space who looks to the Earth. The Atlantic streamer starts with a tongue-like structure reaching from Mexico over the Atlantic to Morocco. Later on December, 4 2015 a narrow ozone streamer has been formed reaching from Mexico to Central Europe. The ozone streamer is moved southward and fades away in Figure 3c). The SD-WACCM simulation of the formation and the decay of the Atlantic
streamer is confirmed by the Aura/MLS observations in Figure 3d), e) and f). Please note that the ozone field of Aura/MLS was not used for nudging of the SD-WACCM model run. The structures of the Atlantic streamer in the pure Aura/MLS ozone fields are quite similar as in the SD-WACCM ozone fields. This is a nice confirmation for the nonlinear wave-mean flow interactions in the stratosphere as simulated by SD-WACCM. Generally, the streamer is clearer in the SD-WACCM simulation than in the Aura/MLS observations. There are at least two reasons which may explain this result. The SD-WACCM model simulation
does not resolve all inertia-gravity waves which may disturb the formation and duration of streamers and filaments. Secondly, the limited horizontal and temporal resolution of the Aura/MLS observations may render a clear detection of streamers and filaments.

    Figure 4 zooms into the Atlantic streamer on December 4, 2015. The colour shading gives the ozone value and the arrows depict the horizontal wind vector. The largest arrows correspond to wind speeds of about 100 m/s. The figure clearly shows
that a narrow stream of ozone-rich air comes over the Atlantic to France, and it turns southward over East Europe.

    Figure 5a) utilizes water vapour as a tracer of polar vortex air and shows the spatial distribution of stratospheric water vapour at 34 km altitude on December 4, 2015. Small arrows are indicating the horizontal wind. An erosion region of polar, water vapour rich air appears above the Caspian Sea ending in a long filament of water vapour pointing in south-west direction. This finding is in a qualitative agreement with the vortex filamentation studies of Müller et al. (2003) and Koh and Legras (2002).



Figure 5b) shows the spatial distribution of stratospheric ozone at 34 km altitude on December 4, 2015. Comparison with Figure 5a) shows that the Atlantic ozone streamer is located in the edge region of the polar vortex. In addition the ozone streamer turns southward before reaching the vortex erosion region above the Caspian Sea. The ozone streamer, the erosion region and the water vapour filament can be regarded as the cat-eye of a breaking planetary wave. The SD-WACCM simulation and the special meteorological situation in Figure 5 are appropriate to visualize the surf zone of the mid-latitude stratosphere

in winter (McIntyre and Palmer, 1984). Figure 5a) and b) also show the anti-correlation of the spatial distributions of ozone and water vapour in the mid-stratosphere since stratospheric polar air is rich in water vapour and poor in ozone.

    Finally, we like to compare the water vapour distribution of SD-WACCM (Figure 5a)) with the pure observations of the satellite experiment Aura/MLS on December 4, 2015. Figure 6 shows the result of Aura/MLS at 8.3 hPa which is close to 34 km altitude. The water vapour distributions of Aura/MLS and SD-WACCM are in a good agreement. The vortex erosion region

over the Caspian Sea is unclear in case of Aura/MLS. The water vapour filaments over North Africa and South Pakistan in Figure 6 indicate that there was a transport of water vapour rich air from the polar vortex to the subtropics. The filaments and the vortex erosion are clearer in the SD-WACCM simulation than in the Aura/MLS observations. We suggest that in reality, inertia-gravity waves which are not resolved in the model simulation may disturb the formation and duration of streamers, filaments and vortex erosion regions. Another reason is the limited horizontal resolution of the Aura/MLS limb sounding

observations which is about 200 km at the tangent point.

## 4   Conclusions

An Atlantic streamer was detected in stratospheric ozone observations of the space-based microwave radiometer Aura/MLS and the ground-based microwave radiometers GROMOS and SOMORA in Switzerland. These observations were compared to SD-WACCM simulation data. The timing of the streamer event on December 4, 2015 and the global structure of the Atlantic

streamer agree well for Aura/MLS, SD-WACCM and GROMOS. One can nicely see the extension of the tongue-like structure which transports subtropical ozone-rich air from Mexico to Central Europe. The Atlantic streamer is strongest at altitudes between 30 and 45 km. Eastward wind speeds of about 100 m/s are reached inside the narrow streamer.

    The SD-WACCM simulation of the spatial distributions of horizontal wind, water vapour and ozone in Figure 5 show details of planetary wave breaking in the surf zone at 34 km altitude at northern mid-latitudes on December 4, 2015. The Atlantic

ozone streamer flows eastward in the edge region of the polar vortex. The ozone streamer turns southward before reaching the Caspian Sea where a vortex erosion region is located. The vortex erosion region shows an increase of water vapour rich polar air. A water vapour filament flows from this region in southwest direction. Generally, the spatial distributions of water vapour and ozone are anti-correlated so that the ozone streamer contains water vapour poor air and the water vapour filament contains ozone poor air. The SD-WACCM simulation shows that the Atlantic streamer is a part of the planetary wave breaking process

in the surf zone of the mid-latitude stratosphere in winter. This result is in agreement with Waugh (1996) who reported that transport out of the tropics occurs in Rossby wave breaking events in which filaments of tropical air are drawn into middle latitudes in the winter season. The streamers and filaments are clearer in the SD-WACCM simulation than in the Aura/MLS





observations. We suggest that in reality, inertia-gravity waves which are not resolved in the model simulation may disturb the formation and duration of streamers and filaments. Another reason is the limited horizontal resolution of the Aura/MLS limb sounding observations which is about 200 km at the tangent point.

## 5   Code availability

5    Routines for data analysis and visualization are available upon request by Klemens Hocke.

## 6   Data availability

The ground-based ozone measurements of SOMORA and GROMOS are available in the data centre of the Network for the Detection of Atmospheric Composition Change (http://www.ndacc.org). The Aura/MLS level2 data are available at the Aura Validation Data Center (http://avdc.gsfc.nasa.gov/). The SD-WACCM simulation data of winter 2015/2016 is available by the
10    author Franziska Schranz.

*Author contributions.* Franziska Schranz performed the SD-WACCM model simulation. Klemens Hocke carried out the plots. Eliane Maillard Barras took care on the SOMORA data. All authors contributed to the interpretation of the data sets.

*Acknowledgements.* We thank the Aura/MLS team and NASA/JPL for the microwave limb sounding measurements and the provision of the level2 data set at the Aura Validation Data Center (http://avdc.gsfc.nasa.gov/). We are grateful to the National Center for Atmospheric
15    Research (Boulder) for providing the SD-WACCM model. The study was supported by Swiss National Science Foundation under grant number 200020-160048 and 200021-165516.





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

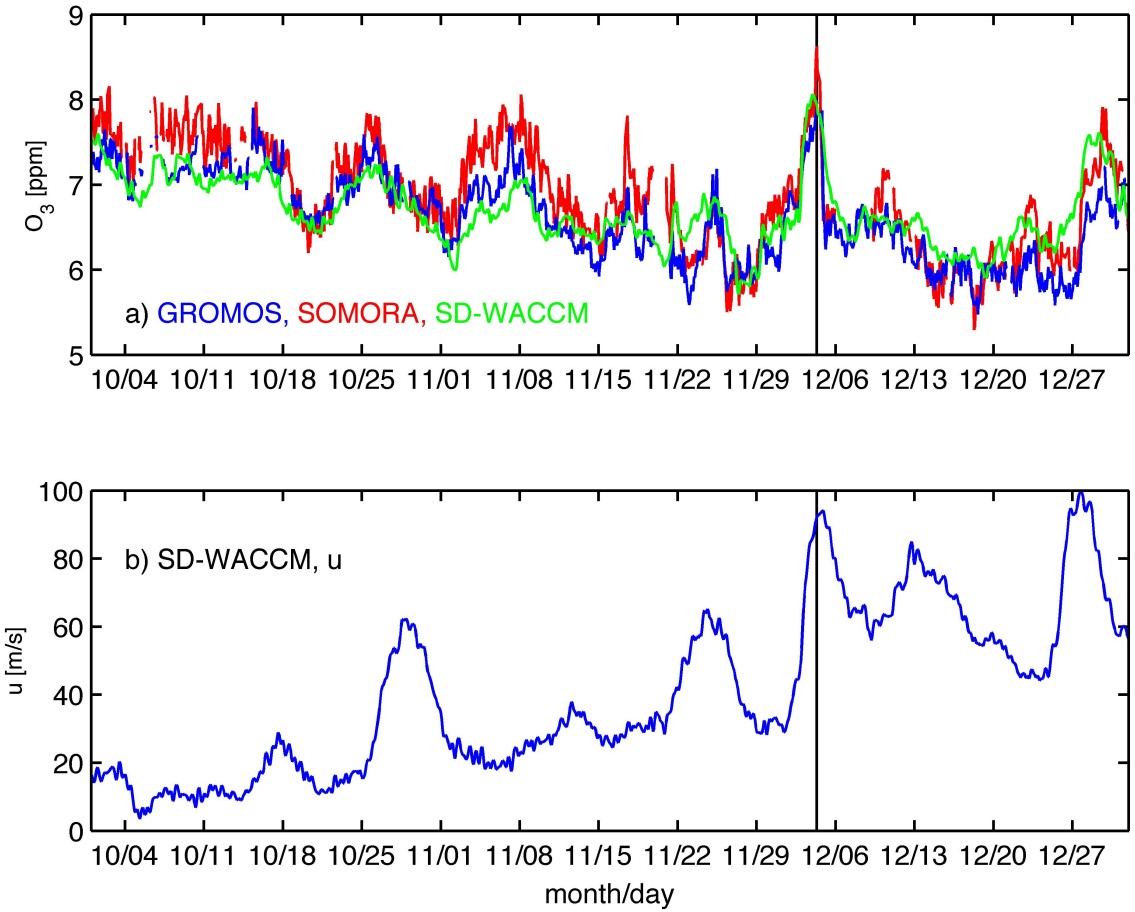

**Figure 1.** Time series of ozone volume mixing ratio and zonal wind at 34 km altitude above Bern, Switzerland from October to December 2015. The vertical red line is at 2015-12-04 12:00 UT when the ozone streamer reached Bern. a) Time series of ozone observed by GROMOS (Bern) and SOMORA (Payerne) versus the simulated SD-WACCM ozone series at the grid point nearest to Bern. b) Time series of eastward wind simulated by SD-WACCM at 34 km altitude.





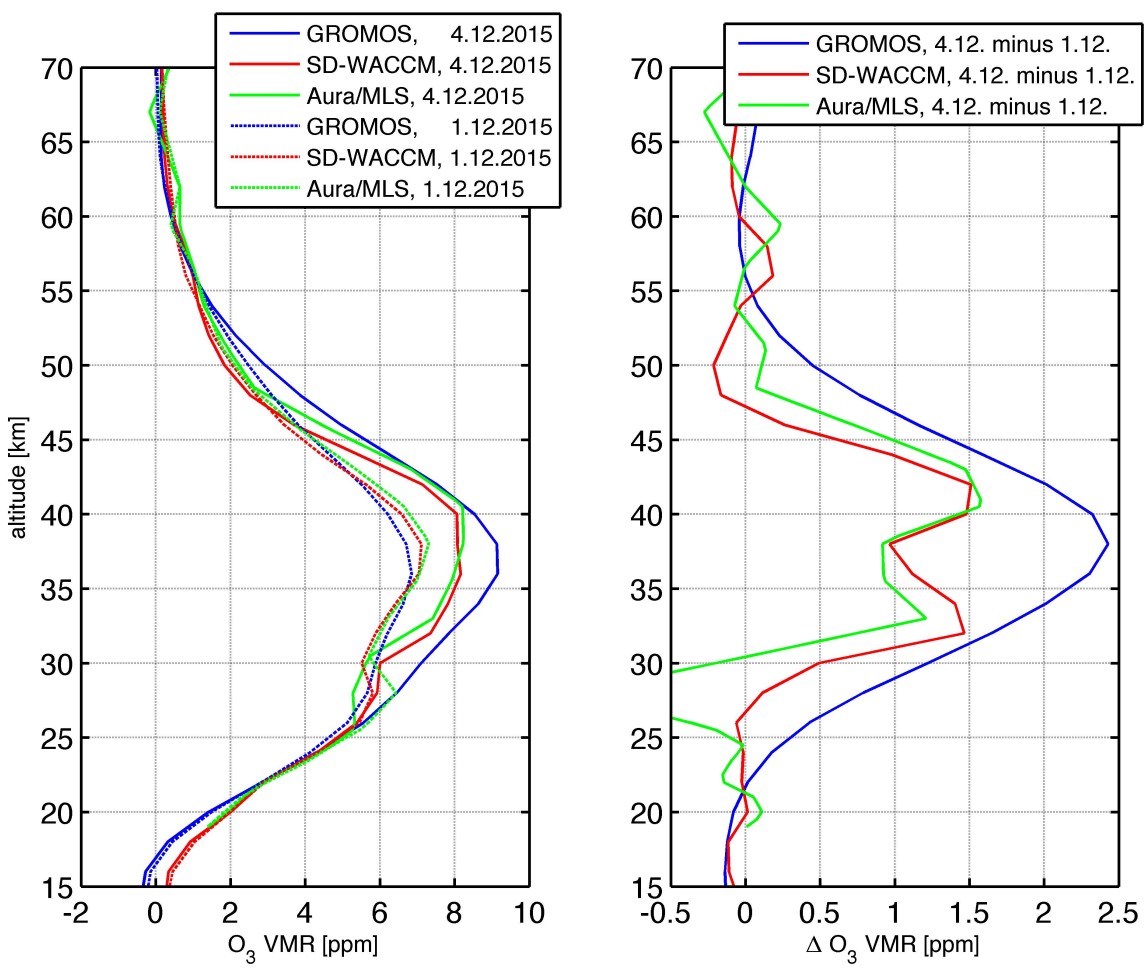

**Figure 2.** Left-hand-side: Vertical ozone profiles at Bern (or close to Bern) before the streamer arrival on 2015-12-01 12:00 UT (dashed line) and at the streamer arrival on 2015-12-04 12:00 UT. The GROMOS observations are indicated by the blue lines, the SD-WACCM results are given by the red lines, and Aura/MLS is shown by the green lines. Right-hand-side: Difference of the ozone profiles from 2015-12-04 and 2015-12-01. The relative uncertainties of GROMOS, SOMORA and Aura/MLS are about 10%.





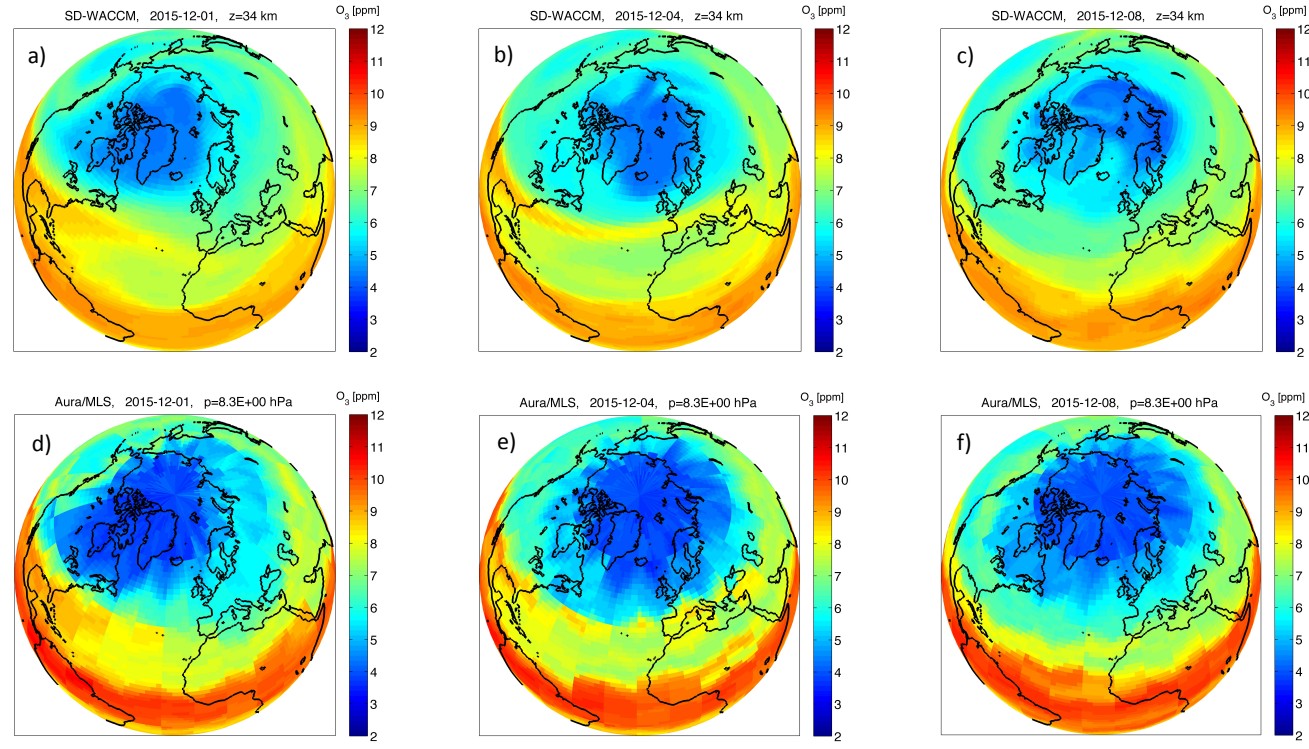

**Figure 3.** a) Begin of an ozone streamer extending from Mexico over the Atlantic to Morocco on 2015-12-01 at 34 km altitude and simulated by SD-WACCM. b) The ozone streamer narrows and extends to Central Europe on 2015-12-04. c) The ozone streamer is shifted southward and fades away on 2015-12-08. The viewgraphs d), e) and f) are based on all valid ozone profiles of Aura/MLS measured during the days 2015-12-01, 2015-12-04 and 2015-12-08.





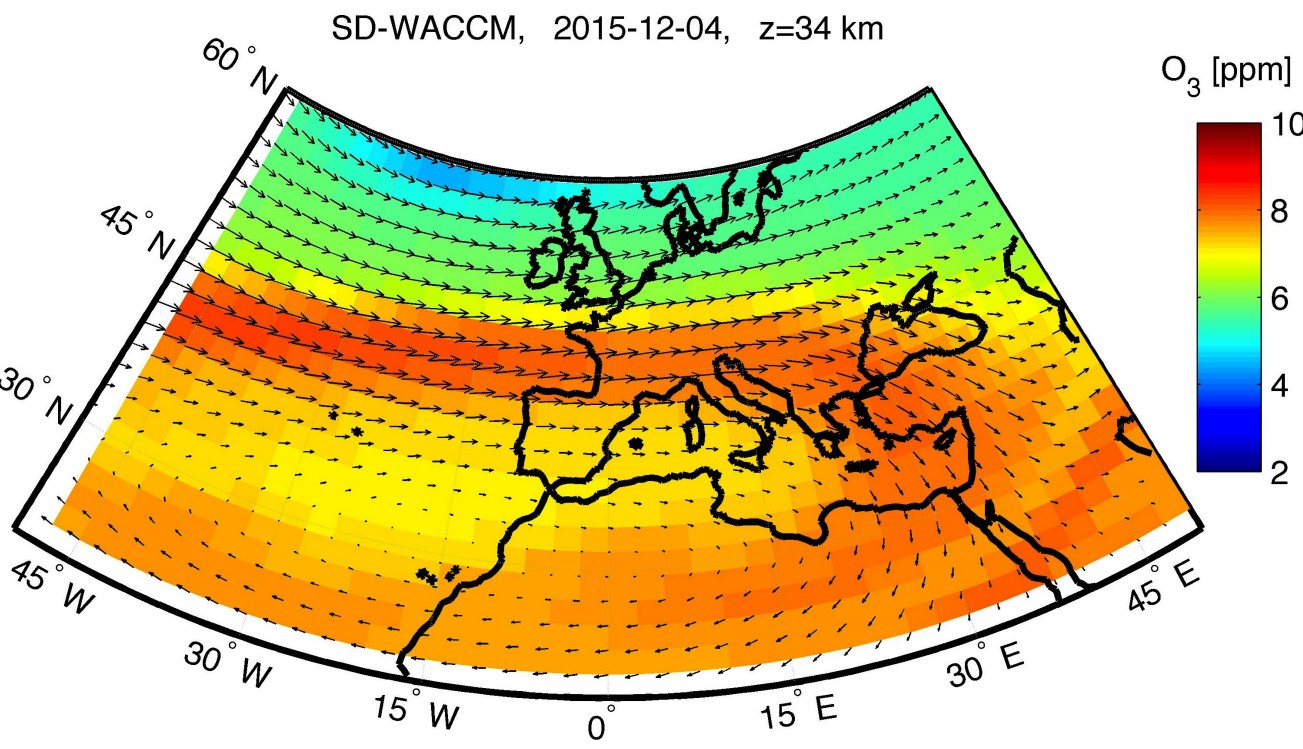

**Figure 4.** The Atlantic ozone streamer reaches Central Europe and turns southward over East Europe. The largest arrows correspond to wind speeds of about 100 m/s within the Atlantic streamer at an altitude of 34 km.



**Figure 5.** Tracer distributions of water vapour (a)) and ozone (b)) in 34 km altitude on December 4, 2015. Water vapour is a tracer of polar vortex air. It indicates an erosion region of the polar vortex located above Caspian Sea in a). Further, a water vapour filament is visible outgoing from the erosion region of the vortex. The ozone distribution in b) is anti-correlated to the water vapour distribution in a). The ozone tracer shows the Atlantic streamer which can be regarded as the outer part of the cat eye of the planetary wave breaking process.





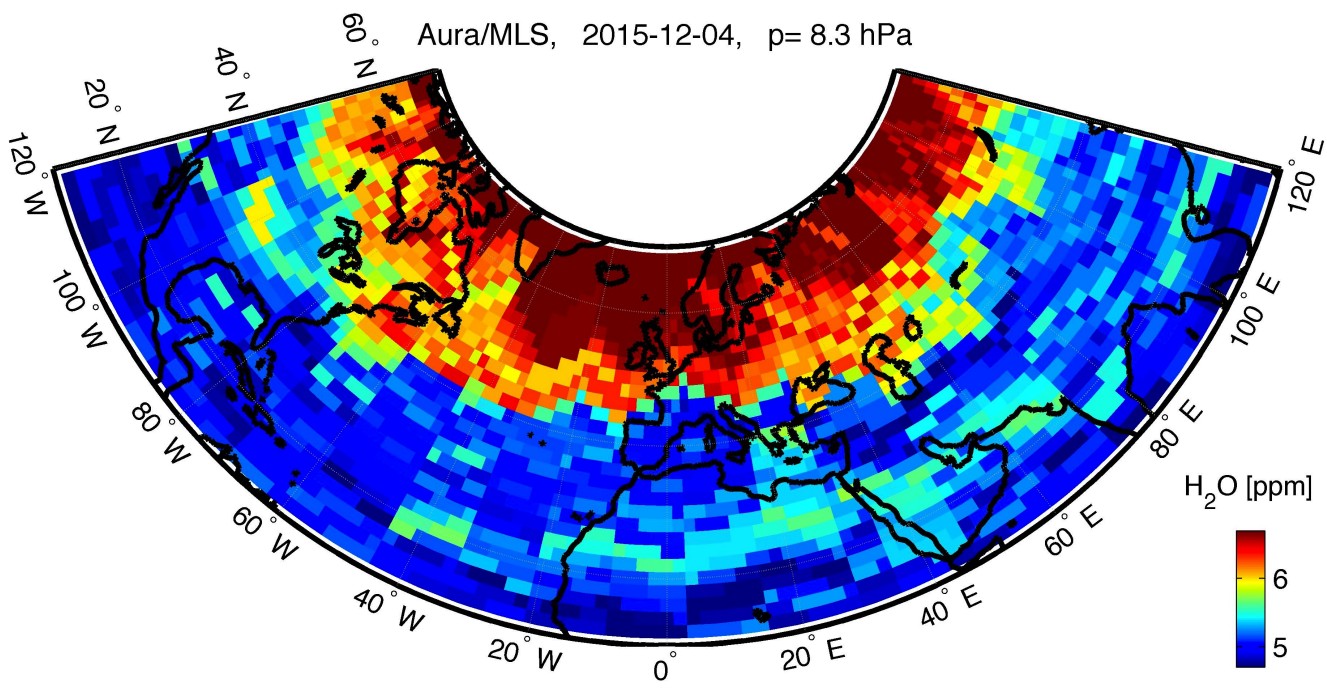

**Figure 6.** Tracer distribution of water vapour at 8.3 hPa (ca. 34 km altitude) observed by Aura/MLS on December 4, 2015. There are H2O filaments or water vapour enhancements above Northern Africa and South Pakistan which is in a rough agreement with Figure 5a).