# Peer review of "An Atlantic streamer in stratospheric ozone observations and SD-WACCM simulation data"

_Atmospheric Chemistry and Physics, 2016_

## Referee Comment (RC1) · Anonymous Referee #2 · 28 Dec 2016

RECOMMENDATION: minor revision

SUMMARY STATEMENT: The paper describes a model-observation case study of an Atlantic streamer. It identifies an ozone streamer, a water vapor filament and the related polar vortex erosion region in different datasets. An extended discussion would complete the paper - I suggest a minor revision.

MAJOR COMMENTS

The discussion should be extended: (1) One topic I could imagine is the represantativity of the observed streamer. How typical is it in relation to the climatologies of streamers (Martius et al., 2007) and Rossby wave breaking (Zülicke & Peters, 2008)? (2) Another topic is the proper resolution of three-dimensional structures in the data. You give some reason for the different appearance of filaments in SD-WACCM vs Aura-

[Figure]

MLS (gravity waves, resolution). You also mention the double-peaked anomaly in SD-WACCM and Aura-MLS vs GROMOS but do not further discuss these differences (10 % at about 40 km). (3) Another point worth a discussion is how close the SD-WACCM ozone concentrations are to the observations. Is it perhaps related to the dynamically produced tracer patterns? I suggest to state the problem of proper sampling of streamers and filaments in the introduction and to discuss them in the last section. For examle you give three times the same reasons for differences in structures from SD-WACCM and Aura-MLS (page 4 line 24, page 5 line 12 and 32) which should be placed in discussion section.

In was also expecting some concluding words on the result of the SD-WACCM validation. My impression after reading was that both the vertical and horizontal resolution is as good as the satellite observations, and also the ozone concentrations are realistic. Some discrepancies to GROMOS remain which should be commented. However, such statements should repeat the spatial and temporal scales of the validation exercise.

MINOR COMMENTS

page 2, line 10: "its included" is possibly not the best wording - may be "included" is better to read

page 2, line 27: The height ranges do not correspond to fig. 3: there it goes from 15 to 70 km, but here you write of profiles from 25 to 70 km?

page 2, line 28: replace "with an" with "to"

page 4, lines 1-3: Please check the reference to Fig. 1a and 1b! I think, in line 1 you refer to 1a and ind line 3 to 1b?

page 4, line 1: Replace "shown" with "included"

page 4, line 2: Replace "sees" with "reproduces"

page 4, line 22: Delete "nice" which sounds too subjective.

page 4, line 30: Replace "comes" with "extends"

page 5, line 7: Delete "pure"

page 5, line 9: What do you mean with "good agreement" - the overall zonal structure or the orders of magnitude? Please, specify.

page 5, line 16: This section does not only contain the conclusions if any, it is more a summary and repeated discussion. I suggest to place discussion and conclusion here, while a summary for this short paper is not necessary.

page 5, line 20: Delete "nicely"

page 5, line 31: You always refered to the high-ozone subtropical structures a "streamers" - may be its better to use this term here instead of "filaments"?

REFERENCES

Martius, O. C., C. Schwierz & H. C. Davies, 2007: Breaking waves at the tropopause in the wintertime Northern Hemisphere: Climatological analyses of the orientation and the theoretical LC1/2 classification. J. Atmos. Sci. 64: 2567 - 2592. doi:10.1175/JAS3977.1.

Zülicke, C. & D. H. W. Peters, 2008: Parameterization of strong stratospheric inertia-gravity waves forced by poleward breaking Rossby waves. Mon. Wea. Rev. 136, 1: 98 - 119. doi:10.1175/2007MWR2060.1.

---

## Referee Comment (RC2) · Anonymous Referee #1 · 28 Dec 2016

The manuscript presents a case study of an Atlantic streamer, which was observed by MLS and ground based remote sensing measurements of ozone and simulated in a specified dynamics WACCM model run. The study is well written and fits the scope of ACP. Although it does not seem very remarkably to me that such a streamer can be observed and modeled, there are not many detailed descriptions of such events in the literature, yet. Therefore the manuscript represents a substantial contribution to the scientific progress. Major points

My major concern is, how the temporal and spatial sampling of the different data sets is treated and discussed:

1) Temporal sampling of MLS and SD-WACCM data

From the description of the model and MLS data I don't understand what data for a

certain date (as shown in Fig. 3) actually means?

MLS is on a sun synchronous orbit but can measure both night and day, so within one day there are usually measurements with small spatial difference but with 12h temporal difference. Do you take both night and day measurements into account? Considering the movement of the streamer shown in Fig. 3 I could imagine, that this can contribute to the less clear appearance of the streamer in the MLS data? Maybe it is better to use only one of them?

What is one day for the SD-WACCM data? Usually models run on smaller time steps (but probably do not save the data at each model time steps) – is one date as shown in Fig. 3 one certain time (e.g. 00:00 or 12:00 UTC)? Or an average over all/certain times of one day? For some models, there exist data which are actually sampled to correspond to the measurement times of satellites, see e.g. Joeckel et al., 2010. If such a data set exists for SD-WACCM and MLS this would be ideal to identify to what extend the differences are caused by the sampling.

2) Spatial sampling of the MLS data

On page 4 line 26, page 5 line 14/15 and page 6 line 3 and you underline the issue with the horizontal resolution (about 200km) of the limb sounding data from MLS. This can cause problems to resolve structures but I would assume that the horizontal sampling (about 165km if all profiles are valid) has a similar effect? As mentioned on page 4 line 26 also the temporal resolution (see (1)) can play a role?

3) Selection of a height or pressure level for comparison

For one location one can chose the "nearest neighbor" pressure level to one altitude as described on page 5 line 8, but for a larger region this can cause differences. In Fig. 3 there is a higher zonal O3 gradient in the MLS data, maybe this is caused by using pressure instead of height in km? I think it would be better for Fig. 3 and 5/6 to use the same vertical coordinate. (For MLS there is a geopotential height field which could be

used to calculated geometric height and for model data pressure should be available as well?)

Minor points

Section 2.2

Filtering of the MLS data: together with the MLS Level 2 data sets usually a manual is distributed, how valid data should be selected. Since the authors write they use "all valid ozone profiles" I assume that these criteria were applied? For ozone and for water vapour (as far as I remember the data there could be different profiles valid for water vapour than for ozone)? I think it would be good if these criteria are explained or at least that the data quality document (Livesey et al., 2016) is cited.

Choice of the figures

I think there is to much redundant information in Figure 4 and 5b. Therefore I would suggest to combine 5b and 4 into one to make it clear that 4 is a zoom into 5b. At the same time I think it would be better to combine Figure 5a and 6 into one to make it easier for the reader to compare them.

Literature

Jöckel, P., Kerkweg, A., Pozzer, A., Sander, R., Tost, H., Riede, H., Baumgaertner, A., Gromov, S., and Kern, B.: Development cycle 2 of the Modular Earth Submodel System (MESSy2), Geosci. Model Dev., 3, 717-752, doi:10.5194/gmd-3-717-2010, 2010.

Nathaniel J. Livesey, William G. Read, Paul A. Wagner, Lucien Froidevaux, Alyn Lambert, Gloria L. Manney, Luis F. Millan Valle, Hugh C. Pumphrey, Michelle L. Santee, Michael J. Schwartz, Shuhui Wang, Ryan A. Fuller, Robert F. Jarnot, Brian W. Knosp, Elmain Martinez: "Version 4.2x Level 2 data quality and description document.", http://mls.jpl.nasa.gov/data/v4-2_data_quality_document.pdf

---

## Author Comment (AC1) · 15 Feb 2017

Dear Editor, dear Reviewers,

Please find our response in the attached pdf file.

Thank you and best regards, Klemens Hocke

Please also note the supplement to this comment:
http://www.atmos-chem-phys-discuss.net/acp-2016-996/acp-2016-996-AC1-supplement.pdf

---

## Author Response (AR2)

**Dear Editor, Dear Reviewers,**

**We included your latest minor corrections in the second minor revision of 1 March 2017.**

**Thank you**
**Klemens Hocke**

**------------------------------------------------------------------------**

**Dear Editor, Dear Reviewers,**

We thank you for the constructive review process. Considering your comments, we performed a moderate revision. The four main changes are:

1) Intercomparisons in Figures 3-5 are now all on the 8.3 hPa-level. Particularly the meridional ozone gradient in Figure 3 is now in a better agreement for SD-WACCM and Aura/MLS.

2) We re-arranged the Figures 4 and 5 so that the intercomparison of water vapour from SD-WACCM and Aura/MLS becomes easier in Figure 5.

3) We extended the discussion and we added a discussion section. The new reference Polvani and Saravanan (2000) helps us to understand the shape of the polar vortex when planetary wave breaking occurs.

4) We added sentences about the spatio-temporal resolution of Aura/MLS maps. We tried to reduce the temporal interval of the Aura/MLS map by taking only daytime data around noon but there was no positive effect. Thus we take 24 hour-intervals of Aura/MLS data centered at 12:00 UTC of the selected day.

**Point-to-point response to Reviewer 1:**

1) Temporal sampling of MLS and SD-WACCM data From the description of the model and MLS data I don't understand what data for a certain date (as shown in Fig. 3) actually means?
MLS is on a sun synchronous orbit but can measure both night and day, so within one day there are usually measurements with small spatial difference but with 12h temporal difference. Do you take both night and day measurements into account? Considering the movement of the streamer shown in Fig. 3 I could imagine, that this can contribute to the less clear appearance of the streamer in the MLS data? Maybe it is better to use only one of them?
What is one day for the SD-WACCM data? Usually models run on smaller time steps (but probably do not save the data at each model time steps) – is one date as shown in Fig. 3 one certain time (e.g. 00:00 or 12:00 UTC)? Or an average over all/certain times of one day? For some models, there exist data which are actually sampled to correspond to the measurement times of satellites, see e.g. Joeckel et al., 2010. If such a data set exists for SD-WACCM and MLS this would be ideal to identify to what extend the differences are caused by the sampling.

Data for a certain date are coming from a 24 hour interval centered at 12:00 UTC of the selected day. Yes, it is day and nighttime data. Because of your comment we produced an ozone map from a 12 hour interval centered at 12:00 UTC but the result was almost the same since there is no strong diurnal ozone variation at 8.3 hPa. The time resolution of the SD-WACCM output data is 2 hours. We added this important information now. Actually, one could run the model with a higher temporal resolution too.

2) Spatial sampling of the MLS data
On page 4 line 26, page 5 line 14/15 and page 6 line 3 and you underline the issue with the horizontal resolution (about 200km) of the limb sounding data from MLS. This can cause problems to resolve structures but I would assume that the horizontal sampling (about 165km if all profiles are valid) has a similar effect? As mentioned on page 4 line 26 also the temporal resolution (see (1)) can play a role?

In the revised manuscript we inform that the "poor" resolution in longitude of Aura/MLS (ca. 24 deg) is possibly a limiting factor. Nevertheless there are other days in the Aura data center which show a connection between vortex erosion region and outgoing water vapour filament. Actually, Aura/MLS is the limb sounder with the most dense sampling of the atmosphere.

3) Selection of a height or pressure level for comparison
For one location one can chose the "nearest neighbor" pressure level to one altitude as described on page 5 line 8, but for a larger region this can cause differences. In Fig. 3 there is a higher zonal O3 gradient in the MLS data, maybe this is caused by using pressure instead of height in km? I think it would be better for Fig. 3 and 5/6 to use the same vertical coordinate. (For MLS there is a geopotential height field which could be used to calculated geometric height and for model data pressure should be available as well?)

Yes, we agree. Now we transfered all the maps of Figures 3-5 to the pressure level 8.3 hPa. The agreement of SD WACCM and Aura/MLS is improved in the global maps of Figure 3 as you assumed.

Minor points
Section 2.2
Filtering of the MLS data: together with the MLS Level 2 data sets usually a manual is distributed, how valid data should be selected. Since the authors write they use "all valid ozone profiles" I assume that these criteria were applied? For ozone and for water vapour (as far as I remember the data there could be different profiles valid for water vapour than for ozone)? I think it would be good if these criteria are explained or at least that the data quality document (Livesey et al., 2016) is cited.

Now, we cite the technical report of Livesey et al. (2016). Our read program considers the changing validity-thresholds of different species.

I think there is to much redundant information in Figure 4 and 5b. Therefore I would suggest to combine 5b and 4 into one to make it clear that 4 is a zoom into 5b. At the same time I think it would be better to combine Figure 5a and 6 into one to make it easier for the reader to compare them.

Okay, we changed and combined the figures acccording to your idea.

**Point-to-point response to Reviewer 2:**

The discussion should be extended: (1) One topic I could imagine is the represantativity of the observed streamer. How typical is it in relation to the climatologies of streamers (Martius et al., 2007) and Rossby wave breaking (Zülicke & Peters, 2008)?

We added a discussion section where we use the Zülicke & Peters (2008) in order to emphasize the connection between planetary wave breaking and the generation of inertia-gravity waves. The study by Martius et al. (2007) is not so useful for our purpose since it is about planetary wave breaking at the tropopause. Planetary wave breaking in the UTLS region is quite different to planetary wave breaking in the middle and upper stratosphere. Here, we found that the study by Polvani and Saravanan (2000) is valuable to understand the "comma-shape" of the polar vortex during planetary wave breaking. The represantativity of the observed streamer was investigated in Krüger et al. (2005). The Atlantic streamer is quite typical since the middle stratospheric polar vortex is often shifted by a zonal wavenumber-1 forcing to the European longitude sector.

(2) Another topic is the proper resolution of three-dimensional structures in the data. You give some reason for the different appearance of filaments in SD-WACCM vs Aura-MLS (gravity waves, resolution). You also mention the double-peaked anomaly in SD- WACCM and Aura-MLS vs GROMOS but do not further discuss these differences (10 % at about 40 km).

In the text, we explain that the vertical resolution of GROMOS is not good enough to resolve the double peak, instead of the double peak GROMOS measures a single peak located between the double peaks of Aura/MLS and SD-WACCM. Vertical smoothing of the double peak-profiles would result into a smooth single peak profile. Thus there is no contradiction between GROMOS and Aura/MLS or SD-WACCM. In the revised manuscript we added more information on the horizontal resolution of Aura/MLS.

(3) Another point worth a discussion is how close the SD-WACCM ozone concentrations are to the observations. Is it perhaps related to the dynamically produced tracer patterns? I suggest to state the problem of proper sampling of stream- ers and filaments in the introduction and to discuss them in the last section. For examle you give three times the same reasons for differences in structures from SD-WACCM and Aura-MLS (page 4 line 24, page 5 line 12 and 32) which should be

Now we explain that the longitude resolution is about 24 deg. This is the spacing between two subsequent orbits of Aura/MLS. On the other hand, the latitude resolution is better (1.5 degrees). The integration time of the Aura maps is 24 hours. We tried a shorter integration time of 12 hours (only daytime data) for the water vapour map  but the result was not better. In summary it remains open if Aura/MLS should have seen the vortex erosion region in conjunction with the water vapour filament. We guess it is most likely due to both: small-scale variability not resolved by the model and a coarse longitude resolution of Aura/MLS.  However, there are other days where Aura/MLS observes the comma-shape of the disturbed polar vortex together with an outgoing water vapour filament - but then the Atlantic ozone streamer is not above Switzerland.

In was also expecting some concluding words on the result of the SD-WACCM valida- tion. My impression after reading was that both the vertical and horizontal resolution is as good as the satellite observations, and also the ozone concentrations are realistic.

We agree that SD-WACCM makes a very good job to simulate the ozone maps which are observed by Aura/MLS and to simulate the ozone time series which are observed by SOMORA and GROMOS. The agreement in Figure 3 is improved now since we have put all the maps to the 8.3 hPa level (and not some maps to z=34 km). We add a concluding remark on the result of the SD-WACCM validation.

Minor comments: ...

Thank you for the minor comments which helped us to improve the style of the article!